# Current Insights and Future Directions in Scar Management and Skin Regeneration

**DOI:** 10.3390/ijms262110636

**Published:** 2025-10-31

**Authors:** Dominika Szlachcikowska, Katarzyna Mazurek, Monika Magiera, Grzegorz Jama, Anna Tabęcka-Łonczyńska

**Affiliations:** 1Department of Biotechnology and Cell Biology, Medical College, University of Information Technology and Management in Rzeszow, Sucharskiego 2, 35-225 Rzeszow, Poland; dszlachcikowska@wsiz.edu.pl; 2Health and Beauty Clinic, Individual Specialist Nursing Practice, Lubelska 13, 22-400 Zamość, Poland; kmazurek2302@gmail.com; 3Monika Magiera, Beauty Expert, Krasne 32a/31, 36-007 Krasne, Poland; monikamagieramakeup@gmail.com; 4Department of Interventional Radiology and Neuroradiology, University Clinical Hospital No. 4, Jaczewskiego 8, 20-954 Lublin, Poland; grzegorzjama@gmail.com

**Keywords:** tropocollagen, skin regeneration, wound healing, scar, dermatology

## Abstract

Skin scarring, including hypertrophic scars and keloids, often results from dysregulated collagen deposition during wound healing. Tropocollagen (TC), the soluble triple-helical precursor of collagen fibers, serves as the fundamental structural unit of the extracellular matrix (ECM) and plays a pivotal role in tissue repair. This review summarizes current knowledge on collagen and TC in wound healing, scar management, and regenerative dermatology. TC self-assembles into fibrils, providing structural support, while interacting with fibroblasts and growth factors such as transforming growth factor beta (TGF-β) and vascular endothelial growth factor (VEGF) to regulate ECM remodeling, angiogenesis, and tissue regeneration. Various collagen preparations, including hydrolyzed collagen, gelatin, and native fibrillar forms, differ in molecular structure, bioavailability, and therapeutic applications. Emerging strategies, including collagen- and TC-based hydrogels, nanomaterial composites, and smart wound dressings, enhance stability, targeted delivery, and clinical efficacy. Despite promising preclinical and early clinical data, standardized preparations and robust randomized trials are needed to validate TC’s therapeutic potential and optimize its application in scar prevention and wound repair.

## 1. Tropocollagen in Skin Regeneration and Scar Modulation

Skin scarring represents a natural stage of the wound healing process. However, in many cases, it leads to unfavorable aesthetic or functional outcomes, such as hypertrophic scars or keloids [1,2]. These alterations are primarily driven by dysregulation in the deposition of collagen, particularly types I and III, which determine the structure and mechanical properties of the newly formed tissue [3,4]. Collagen, in the form of tropocollagen (TC)—the fundamental structural unit of collagen fibers—plays a pivotal role in the remodeling of the extracellular matrix (ECM) [5]. TC is formed through the triple helical assembly of polypeptide chains, which subsequently organize into higher-order fibrillar structures, providing tissues with tensile strength and elasticity [6]. For this reason, in recent years, TC has attracted considerable attention as a natural and essential structural component of collagen, increasingly explored for applications in wound and scar management [7].

In dermatology and skin surgery, TC has gained particular attention as a component of biomaterials supporting wound repair [8]. Preclinical studies indicate that the administration of collagen in the form of gels, dressings, or scaffolds influences fibroblast activity, modulates growth factors such as transforming growth factor beta (TGF-β), and regulates matrix metalloproteinases (MMPs), which govern the balance between ECM deposition and degradation [8,9]. Such activity promotes a more orderly remodeling of scar tissue and may reduce the risk of pathological scar formation [9].

Clinical applications of collagen-based preparations, including those derived from fish and bovine sources, are well-documented in the treatment of chronic wounds, such as venous ulcers [10]. Randomized clinical trials have demonstrated that the topical use of collagen gels accelerates wound closure and improves the quality of granulation tissue [10]. Although most studies focus on chronic wounds, many of the underlying mechanisms—particularly those related to fibroblast regulation and angiogenesis—are directly relevant to surgical wound healing and the prevention of abnormal scar formation [8,9].

Nevertheless, a clear research gap remains regarding the specific effects of TC on cutaneous scarring. Large-scale randomized clinical trials evaluating its efficacy in preventing hypertrophic or acne-related scars are still lacking. The development of TC-based biomaterials, including hydrogels and nanotechnology-modified scaffolds, opens new perspectives in regenerative and aesthetic dermatology [6,8]. The aim of the current review is to summarize the current knowledge on TC in the context of skin regeneration and scar healing, highlighting both established clinical applications and areas requiring further investigation.

## 2. Brief Characterization of Collagen and Tropocollagen

Collagen is a key structural protein of the ECM, serving as the main scaffold that determines the functioning and organization of cells within tissues [11,12]. It is characterized by its high tensile strength, which is essential for the structure of skin, bones, cartilage, and other connective tissues [13,14]. Collagen also exhibits biocompatibility, biodegradability, and low immunogenicity, making it an ideal material for tissue engineering, regenerative medicine, wound healing, orthopedics, cardiology, and dermatology [15,16,17,18,19,20,21].

The fundamental structural unit of collagen is TC—a semi-elastic molecule approximately 300 nm in length and 1.5 nm in diameter [22]. It is composed of three polypeptide chains (α-chains) that are twisted around a right-handed helix, forming a stable, triple-helical structure [22]. Each chain contains approximately 1000 amino acids, with the most frequently recurring sequences composed of glycine, proline, and hydroxyproline [22]. The amino acid composition and the sequence of amino acids in the chain. However, it has been found that types I, II, and III constitute the predominant proportions of total collagen in the human body [3,4]. Figure 1 shows a schematic of collagen assembly and ECM regulation.

## 3. The Role of the Extracellular Matrix (ECM) in the Skin

The extracellular matrix (ECM) is a dynamic and complex network structure that provides both structural and biochemical support to cells and their microenvironment. Through the ECM, tissue integrity is constantly monitored, while cell adhesion, migration, proliferation, and differentiation are regulated. Moreover, it serves as a reservoir for growth factors and cytokines [1,23]. The main component is TC, which assembles into fibrillar networks that provide tissues and organs with specific properties: tensile strength, elasticity, and resistance to mechanical stress [5]. However, for all processes occurring within the ECM—such as wound healing—to proceed properly, accurate regulation of ECM remodeling is essential. Disturbances in collagen deposition or degradation may lead to pathological outcomes, such as hypertrophic scars, keloids, or fibrosis [2,6]. Recent studies indicate that the use of collagen-based biomaterials, including TC gels, scaffolds, and hydrogels, can modulate ECM remodeling, enhance fibroblast activity, and support angiogenesis, thereby promoting proper tissue repair and minimizing the risk of scar formation [2,6,8,9,10]. Understanding and harnessing the dynamic interactions between cells and ECM, therefore, represent a key challenge for the development of advanced regenerative therapies and clinical interventions in dermatology and wound healing.

## 4. TC and ECM Remodeling in Wound Healing

ECM remodeling is critical for tissue repair, dictating the mechanical and functional properties of the regenerated tissue [24]. This phase involves a finely tuned balance between collagen synthesis and degradation, mediated by fibroblasts, matrix metalloproteinases (MMPs), and key growth factors, notably TGF-β [25,26,27]. Following the inflammatory phase, fibroblasts become the key effector cells coordinating with infiltrating immune cells, mainly macrophages and neutrophils, to remodel the provisional matrix. Pro-inflammatory cytokines such as IL-1β and TNF-α, secreted by activated immune cells, stimulate fibroblast proliferation and migration into the wound bed. As inflammation resolves, macrophages undergo polarization toward an M2 phenotype that secretes IL-10 and TGF-β, driving fibroblast differentiation into contractile myofibroblasts [28,29,30]. These myofibroblasts express α-smooth muscle actin (α-SMA) and deposit type I and III collagen, fibronectin, and proteoglycans, gradually replacing the provisional fibrin matrix with a mature ECM.

Within the ECM, TC self-assembles into fibrils, forming a three-dimensional scaffold that provides mechanical support and maintains tissue architecture [2,31]. This structural framework also transmits mechanical cues to resident cells, linking physical forces to cellular responses. Fibroblast adhesion occurs via integrin receptors (primarily α1β1 and α2β1), which bind collagen motifs and activate focal adhesion kinase (FAK), MAPK, and PI3K/Akt signaling cascades [32,33,34]. This mechanotransduction links the biochemical environment with the mechanical properties of the regenerating tissue [35].

As fibroblasts are activated, they synthesize new collagen and other ECM components, contributing to matrix deposition and tissue regeneration [36].

During tissue remodeling, TC is subject to degradation by MMPs, a process that releases growth factors such as TGF-β and vascular endothelial growth factor (VEGF). The proteolytic activity of MMPs (particularly MMP-1, MMP-2, MMP-9, and MMP-13) is tightly regulated by tissue inhibitors of metalloproteinases (TIMPs) to ensure a controlled turnover of ECM components [37,38]. Dysregulation of this MMP/TIMP balance may lead to either excessive degradation or fibrotic accumulation of the matrix [37]. Moreover, collagen-derived peptides generated during MMP-mediated cleavage act as matrikines, bioactive fragments that further modulate fibroblast and endothelial cell behavior, enhancing migration and angiogenesis [39]. These matrikines amplify fibroblast and endothelial cell activation, reinforcing ECM turnover and tissue regeneration. This finely tuned crosstalk creates a positive feedback loop. MMP-mediated ECM degradation releases growth factors such as TGF-β and VEGF, which further stimulate angiogenesis, fibroblast activity, and ECM remodeling, accelerating wound closure and the restoration of tissue integrity [40,41]. Simultaneously, macrophages transition from an M1 (pro-inflammatory) to M2 (pro-regenerative) phenotype, releasing IL-10 and TGF-β, which suppress inflammation and promote granulation tissue formation [42]. The interplay among fibroblasts, immune cells, and endothelial cells establishes a coordinated microenvironment that directs proper tissue reconstruction [43].

Native TC retains its fibrillogenic capacity, which is essential for reconstructing the structural and functional properties of the ECM during wound repair [44]. Proper remodeling yields a thin, elastic, and functional scar, whereas disruptions can result in abnormal collagen deposition, leading to hypertrophic scars or keloids [45]. Overall, the remodeling phase represents a dynamic equilibrium between synthesis, degradation, and cellular signaling, ensuring that matrix architecture and mechanical integrity are restored in a controlled manner [46]. Figure 2 shows TC-fibroblast interactions leading to fibroblast activation, ECM synthesis, and tissue regeneration.

## 5. TC and Other Collagen Forms in Therapeutic Applications

Collagen exists in various forms, each serving distinct roles in therapeutic contexts. This review outlines the molecular structures and functional differences between TC and other collagen derivatives, including hydrolyzed collagen, denatured collagen (gelatin), and native fibrillar collagens (types I, II, and III), Each of these forms is characterized by specific molecular configurations, stability profiles, and modes of biological interaction, which collectively dictate their functional performance in medical and dermatological contexts. The following section describes the principal features of these collagen forms, with particular emphasis on their structural attributes, therapeutic utility, and translational relevance (Table 1 and Table 2).

### 5.1. Hydrolyzed Collagen (Collagen Peptides)

Hydrolyzed collagen, or collagen peptides, results from the enzymatic breakdown of native collagen into smaller peptides, typically ranging from 3 to 6 kDa. This process enhances the bioavailability of collagen-derived amino acids, facilitating their absorption and utilization in the body. Hydrolyzed collagen has been shown to improve skin hydration and elasticity, reduce the appearance of wrinkles, and support joint health by stimulating the synthesis of extracellular matrix components [47,48].

### 5.2. Denatured Collagen (Gelatin)

Denatured collagen, commonly known as gelatin, is produced by subjecting native collagen to thermal or chemical processes that disrupt its triple-helical structure, resulting in a random coil conformation [49,50]. Gelatin retains some functional properties of collagen but lacks the structural integrity necessary for forming fibrils [51]. It is primarily utilized in pharmaceutical and food industries as a gelling agent and in wound dressings due to its biocompatibility [52,53,54,55].

### 5.3. Native Fibrillar Collagens (Types I, II, and III)

Native fibrillar collagens, predominantly types I, II, and III, constitute the principal structural framework of connective tissues [23]. These collagens are defined by their characteristic right-handed triple-helical conformation, which enables supramolecular self-assembly into fibrils stabilized by covalent intermolecular cross-links [2]. Type I collagen is the most abundant isoform in mammals and is primarily distributed in the skin, tendons, and bones, where it confers remarkable tensile strength and structural stability [23]. Type II collagen is the main collagenous component of cartilage, where its fibrillar organization underpins the tissue’s resilience and capacity to withstand compressive forces [23]. Type III collagen, frequently co-localized with type I, is enriched in pliable tissues such as skin, vascular walls, and visceral organs, contributing to elasticity and structural support during tissue remodeling [23]. Owing to these biomechanical and biological properties, native fibrillar collagens are widely exploited in the field of tissue engineering and regenerative medicine as biomimetic scaffolds that promote cellular adhesion, proliferation, and matrix deposition, thereby facilitating tissue repair and regeneration [56].

**Table 1 ijms-26-10636-t001:** Comparison of TC and collagen preparations in therapeutic applications.

Collagen Form	Molecular Structure	Key Applications	Stability	Bioavailability	Therapeutic Effects
**TC**	Triple helix; precursor to fibrils [2]	Tissue engineering, wound healing [7,57]	High [58]	Low [59]	Structural integrity, scaffold formation [7]
**Hydrolyzed Collagen**	Short peptides (3–6 kDa) [60]	Oral supplements, cosmetic products [48]	Moderate [61]	High [60]	Skin elasticity, joint health, and bone density [60,62]
**Denatured Collagen**	Random coil; thermally or chemically treated [63]	Gelling agent, wound dressings [54]	Low [64]	Moderate [65]	Biocompatibility, controlled release [65]
**Native Fibrillar Collagens**	Triple helix; cross-linked fibrils [66]	Tissue scaffolds, regenerative therapies [16]	High [67]	Low [68]	Tissue repair, cellular support [16]

**Table 2 ijms-26-10636-t002:** Comparative structural and biological properties of tropocollagen (TC) and collagen derivatives relevant to wound healing and regenerative therapy.

Feature/Property	TC	Collagen Hydrolysates	Collagen Gels
**Structural characteristics**	Native triple helix; able to form fibrils via supramolecular assembly; stabilized by covalent cross-links [2,31]	Short peptides; partial denaturation of triple helix; no fibrillogenesis [40]	Physically or chemically cross-linked gel network; may partially mimic ECM structure [44]
**Interaction with fibroblasts**	Promotes fibroblast adhesion, proliferation, and migration [36]	Limited fibroblast interaction; depends on peptide length and concentration [40]	Supports fibroblast attachment and spreading; bioactivity depends on gel formulation [41]
**MMP activation**	Degraded by MMPs, releasing growth factors such as TGF-β, promoting tissue remodeling [36]	Limited MMP-mediated activity due to small peptide size [40]	Partial MMP-mediated remodeling possible; dependent on network density [44]
**Angiogenesis stimulation**	Indirect via growth factor release; enhances vascularization in wound sites [36]	Minimal effect; small peptides may have weak pro-angiogenic activity [40]	Can support angiogenesis by providing scaffold for endothelial cell migration [41]
**Therapeutic applications**	Wound healing, tissue engineering scaffolds, regenerative medicine [2,31]	Dietary supplements, joint health, skin support [40]	Wound dressings, tissue regeneration scaffolds, drug delivery systems [41,44]

## 6. Utilization of TC/Collagen in Experimental Research

Experimental models provide complementary insights into the role of TC in wound healing and scar formation. Direct evidence for the therapeutic use of isolated TC in wound-healing models is scarce. Most primary studies evaluate fibrillar collagen, collagen blends, or collagen hydrolysates/peptides, while reports specifically using TC as the experimental agent are largely limited to in vitro fibrillogenesis and biophysical characterizations. The relative paucity of in vivo data on TC per se underscores a gap in the literature and justifies the focus of the present study [69]. Table 3, Table 4 and Table 5 present in vitro, in vivo, and clinical studies demonstrating the beneficial effects of collagen and TC application on wound healing and scar formation.

In vitro systems enable the investigation of molecular and cellular mechanisms, including the effects of TC/collagen on fibroblasts, keratinocytes, and other skin-resident cells [70,71,72,73,74,75,76,77,78,79,80,81] (Table 3). In contrast, animal models allow for the assessment of their impact within the complexity of the whole organism, capturing interactions between cells, extracellular matrix remodeling, and systemic responses during wound repair and scar development [57,72,73,74,79,80,82,83,84,85,86,87,88,89,90,91,92,93,94,95,96,97,98,99,100,101,102,103,104,105,106,107,108,109,110,111,112,113]. Together, these approaches form a critical foundation for understanding the therapeutic potential of TC/collagen in skin regeneration, and a summary of the key experimental findings is provided in Table 4.

Continued research in this field is crucial, as it not only validates the efficacy and safety of TC-based treatments but also contributes to the development of innovative approaches for managing hypertrophic scars, keloids, and chronic wounds [114].

Although direct evidence for the therapeutic use of isolated TC in wound-healing models remains limited, many studies on fish-derived collagen employ acid- or pepsin-soluble preparations, which largely consist of triple-helical molecules structurally equivalent to TC. Thus, these reports provide valuable references when evaluating the biological potential of TC itself [83,85,90,92,94,96,98,100,101,104,106,107,111,115,116,117,118,119].

Clinical studies on the use of collagen and TC in wound healing demonstrate several beneficial effects. They show accelerated wound closure compared to standard dressings or conventional therapy. The quality of scars is improved, with reduced hypertrophic scarring and better skin elasticity at the healed site. Collagen application also lowers the risk of infection and can help reduce patient discomfort. Overall, these treatments are well tolerated, with minimal adverse or allergic reactions reported [115,116,117,118,120,121,122,123]. The data obtained from clinical studies clearly indicate the significant therapeutic potential of collagen and TC in enhancing wound healing and improving scar quality. The summary of clinical study findings is presented in Table 5.

A critical comparison of different collagen forms reveals that both native fibrillar collagen and collagen-derived peptides have specific advantages and limitations depending on their structure, source, and intended application. Native fibrillar collagen, particularly type I of bovine, porcine, or marine origin, provides excellent biocompatibility and mechanical stability, serving as a natural ECM-mimicking scaffold that supports cell adhesion, proliferation, and migration [70,98,115,122,123]. Its native triple-helical architecture facilitates tissue integration and controlled degradation, which is advantageous in wound healing and regenerative applications [76,85,93,99]. However, native fibrils often require cross-linking to achieve sufficient stability [104,116], which may alter their biological properties and reduce bioactivity. Moreover, their relatively low solubility and limited processability can restrict uniform incorporation of bioactive agents or growth factors [73,120].

In contrast, collagen peptides and hydrolysates, derived from partial enzymatic or thermal degradation, exhibit superior solubility, diffusion capacity, and bioavailability. These low-molecular-weight forms are readily absorbed and can modulate cellular signaling by stimulating fibroblast proliferation, angiogenesis, and ECM remodeling both in vitro and in vivo [72,80,82,95]. Numerous studies using fish- or jellyfish-derived collagen peptides have demonstrated enhanced wound closure, re-epithelialization, and collagen deposition, particularly when used in hydrogels or composite scaffolds [57,89,92,94]. Nevertheless, their lack of fibrillar organization limits their mechanical strength and structural support, making them less suitable as stand-alone scaffolds for load-bearing or large-tissue regeneration applications [91,118].

Overall, fibrillar collagens provide structural integrity and biomimetic features critical for scaffold-based tissue engineering. At the same time, collagen peptides act as potent bioactive molecules enhancing cell response and regeneration at the molecular level. Hybrid systems combining both forms—such as peptide-functionalized fibrillar scaffolds or peptide-loaded polymeric matrices—appear to offer synergistic benefits, coupling mechanical stability with enhanced biological activity [84,93,120,124]. Future studies should therefore focus on optimizing the ratio and integration of these collagen forms to achieve a balance between structural functionality and bioactivity in different therapeutic contexts [102,103,104,105,106,107,108,109,110,111,112,113]. However, due to substantial heterogeneity in study designs, collagen source, formulation, and experimental conditions, direct comparisons between individual studies remain limited, which constrains the ability to draw fully consistent conclusions within this review.

**Table 3 ijms-26-10636-t003:** In vitro studies on tropocollagen/collagen in scar healing.

Study	Scar Type	Intervention	Outcome	Reference
In Vitro Studies				
To review and compare different human in vitro skin models applied in the study of wound healing and wound healing disorders	Reconstructed human epidermis (keratinocytes), full-thickness skin equivalents (keratinocytes + fibroblasts), 3D models with immune cells	Type I collagen matrices (often bovine/porcine origin)	Improved scaffold stability and biomimicry; enhanced functional relevance as an alternative to animal models	[70]
Evaluation of a biomimetic bilayer antimicrobial scaffold that mimics skin layers and improves complex wound healing	HaCaT keratinocytes; iPSC-derived endothelial cells	Bilayered scaffold: epidermal film (collagen/chitosan) + dermal collagen–glycosaminoglycan scaffold (cross-linked)	Enhanced cell proliferation and migration; enhanced angiogenic potential; improved scaffold stability and biomimicry	[71]
Test interaction of fibroblasts with cross-linked collagen-elastin scaffold	HDF	Bovine type I collagen	Improved scaffold stability; enhanced cell proliferation and migration	[72]
Effect on fibroblast behavior relevant to wound healing	HDF	EC/PLA/collagen mats loaded with silver sulfadiazine	Maintained cell viability and cytocompatibility; enhanced cell proliferation and migration; antibacterial activity	[73]
Electrospun PCL/PLA Scaffolds Are More Suitable Carriers of Placental Mesenchymal Stromal Cells Than Collagen/Elastin Scaffolds	hAMSCs PMSCs	PCL/PLA scaffolds, collagen/elastin scaffolds (Matriderm), Matrigel-coated scaffolds (bovine type I collagen)	Maintained cell viability; enhanced long-term proliferation and migration; enhanced re-epithelialization and wound closure; enhanced angiogenic potential	[74]
Antibacterial collagen wound dressing	HDF	Bobine type I collagen	Maintained cell viability; enhanced cell proliferation and migration; enhanced extracellular matrix synthesis	[75]
Functionalization of collagen-GAG (glycosaminoglycan) scaffolds with Platelet-Rich Plasma (PRP)	BJ fibroblasts	Bovine type I collagen	Maintained cell viability; enhanced cell proliferation and migration; enhanced re-epithelialization and wound closure	[76]
Scratch wound assay	Hemostatic tests; platelet activation analysis; complement system assays; leukocyte activation analysis	Collagen dressings	Enhanced functional tissue regeneration; enhanced wound stabilization and growth factor release; enhanced angiogenic potential	[77]
Chronic/non-healing wounds (ischemic chronic wound model)	NIH 3T3 HUVEC	Electrospun silk fibroin scaffold incorporating Type I collagen peptides with nitric oxide release	Enhanced cell proliferation and migration; enhanced re-epithelialization and wound closure; enhanced angiogenic potential	[78]
Effect of materials or compounds on cell behavior relevant to wound healing	HaCaT HDF	Collagen/Elastin/PCL scaffold Collagen/PCL scaffold	Enhanced cell proliferation and migration; enhanced extracellular matrix synthesis; enhanced wound closure	[79]
Effectiveness of PCL/collagen wound dressings loaded with insulin-chitosan nanoparticles in accelerating wound healing	HDF	Electrospun PCL/Collagen type I scaffolds loaded with insulin-chitosan nanoparticles	Enhanced cell proliferation and migration; enhanced functional regeneration through sustained bioactive release	[80]
Supporting the proliferation and viability of fibroblasts as key cells involved in wound healing	NIH 3T3	Bovine fibrous collagen type I-based cream	Maintained cell viability; enhanced cell proliferation and migration; enhanced extracellular matrix synthesis	[81]

BJ—normal human foreskin fibroblasts; HaCaT—human adult low calcium high temperature keratinocytes; hAMSCs—human amnion-derived mesenchymal stromal cells; HDF—human dermal fibroblasts; HUVEC—human umbilical vein endothelial cells; NIH 3T3—mouse embryonic fibroblast cell line; PCL—polycaprolactone; PLA—polylactic acid; PMSCs—placental mesenchymal stromal cells.

**Table 4 ijms-26-10636-t004:** In vivo studies on tropocollagen/collagen in scar healing.

Study	Scar Type	Intervention	Outcome	Reference
In Vivo Studies				
Animal Model	Wound/Scar Type	Tropocollagen/ Collagen Preparation	Results/ Observations	Reference
Rats (Wistar)	Full-thickness skin defects excisional skin	Different collagen forms (native, hydrolyzed, semidenatured, commercial gels/creams)	Improved wound healing efficacy; enhanced collagen deposition and ECM remodeling; enhanced fibroblast activity	[82]
Rats (Wistar)	Full-thickness skin wounds	Fish scale tropocollagen peptides	Improved wound healing efficacy; enhanced fibroblast activity; enhanced ECM remodeling	[83]
Rats (Sprague-Dawley) (diabetic wound model)	Full-skin defect wounds	Radiation-crosslinked bilayer bovine collagen type I scaffold	Improved wound healing efficacy; enhanced collagen deposition and angiogenesis; reduced inflammation; maintained biocompatibility	[84]
Rats (Sprague–Dawley)	Full-thickness skin defect	Fish collagen sponge (tilapia)	Maintained biocompatibility; enhanced angiogenesis; enhanced collagen deposition and re-epithelialization	[85]
Rats (Wistar)	Full-thickness wound model	EC/PLA/collagen mats loaded with silver sulfadiazine	Improved wound healing efficacy; enhanced collagen deposition and angiogenesis	[73]
BALB/c mice	Full-thickness skin wounds	Bovine type I collagen	Improved mechanical and degradation properties; improved wound healing potential	[72]
Mice	Full-thickness skin wounds	Electrospun PCL/PLA scaffolds, Collagen/Elastin scaffolds, Matriderm	Improved wound healing efficacy	[74]
Rats (Wistar)	Burn wounds	Rabbit skin collagen hydrogel	Improved wound healing efficacy	[86]
Rats (Wistar)	Third-degree burns	Bovine collagen and zinc oxide	Improved wound healing efficacy	[87]
Rats Mice	Excisional wound Full-thickness wound	Collagen/Elastin/PCL scaffold Collagen/PCL scaffold	Improved wound healing efficacy; enhanced tissue regeneration	[79]
Rats	Full-thickness skin wounds	Collagen type I	Improved wound healing efficacy; enhanced re-epithelialization, collagen deposition, angiogenesis; reduced fibrosis	[88]
New Zealand White rabbit	Burn wounds	Porous sponge scaffold of porcine skin–derived collagen and fish scale–derived collagen	Improved wound healing efficacy; reduced scarring; improved physicochemical wound environment	[90]
Rats	Full-thickness skin wounds	Collagen peptides derived from the jellyfish Rhopilema esculentum	Improved wound healing efficacy; enhanced fibroblast proliferation; enhanced collagen deposition; angiogenesis	[92]
Rats	Full-thickness skin wounds on the dorsum of the rats	Fish collagen	Improved wound healing efficacy; enhanced collagen deposition; re-epithelialization	[119]
New Zealand rabbits	Skin burns	Sponges of carboxymethyl chitosan grafted with collagen peptides	Improved wound healing efficacy; enhanced collagen deposition; epidermal regeneration	[57]
Rats	Full-thickness skin wounds	Pepsin-soluble collagen from the skin of Lophius litulo	Improved wound healing efficacy; enhanced fibroblast proliferation; enhanced collagen deposition	[94]
Rats (Wistar)	Full-thickness excisional and linear incisional wounds	Astaxanthin-incorporated collagen hydrogel film and gentamicin-incorporated collagen hydrogel film	Improved wound healing efficacy; enhanced fibroblast proliferation; enhanced collagen deposition; re-epithelialization; enhanced angiogenesis	[96]
Rats (Sprague–Dawley)	Full-thickness excisional wound	Electrospun PCL/Collagen type I (1:1 w/w) scaffolds loaded with insulin-chitosan nanoparticles	Enhanced tissue regeneration; enhanced collagen deposition	[80]
Dogs and cats	Skin lesions of different etiologies (equivalent to 2nd–3rd degree burns), difficult-to-heal lesions	Collagen type I biomembrane	Improved wound healing efficacy; enhanced angiogenesis	[97]
Rats (Wistar)	Skin burns	Chitosan hydrogel combined with marine peptides from tilapia	Improved wound healing efficacy; enhanced fibroblast proliferation; increased expression of FGF2 and VEGF	[98]
Rats	Full-thickness skin wounds	Chitosan-collagen-alginate composite dressing	Enhanced re-epithelialization and granulation; increased expression of EGF, bFGF, TGF-β, and CD31; enhanced tissue regeneration	[99]
Rats (Sprague-Dawley)	Seawater immersion wounds	Shark-skin collagen sponge with polyurethane film	Improved wound healing efficacy; enhanced re-epithelialization, angiogenesis, and granulation; increased expression of TGF-β and CD31	[100]
Rats (Wistar)	Full-thickness burn wound model	PCL–chitosan nanofibers coated with fish scale-derived collagen type I	Improved wound healing efficacy; enhanced collagen deposition and re-epithelialization; enhanced angiogenesis; enhanced tissue regeneration	[101]
Rabbits	Full-thickness skin wounds	Marine collagen peptides derived from Nile tilapia skin	Improved wound healing efficacy; enhanced fibroblast proliferation; enhanced collagen deposition and re-epithelialization	[102]
Göttingen minipigs	Full-thickness skin wounds	Novel collagen-gelatin fleece	Improved wound healing efficacy; enhanced tissue regeneration	[103]
Mouse (diabetic wound model)	Full-thickness wounds	Atelocollagen hydrogel (protease-sensitive, UV-crosslinked)(tilapia)	Improved wound healing efficacy; enhanced collagen deposition; enhanced angiogenesis	[104]
Rat (Sprague-Dawley)	Full-thickness skin defects	Bilayer dermal substitute composed of a polyurethane membrane + knitted mesh-reinforced collagen-chitosan scaffold	Improved wound healing efficacy; enhanced angiogenesis; enhanced tissue regeneration	[105]
Rats	Full-thickness skin wounds	Fish (mrigal) scale collagen	Improved wound healing efficacy; enhanced fibroblast proliferation; enhanced re-epithelialization;	[106]
Mice	Full-thickness skin wounds	Acid-soluble collagen and its hydrolysates from haddock (Melanogrammus aeglefinus) skin	Improved wound healing efficacy; enhanced collagen deposition and re-epithelialization; enhanced angiogenesis	[107]
Guinea pigs	Full-thickness excisional skin wounds	Chitosan-collagen membrane	Improved wound healing efficacy; enhanced epithelial proliferation	[108]
Mice	Full-thickness round wounds	VitriBand”—a cell-free bandage made of adhesive film, silicone-coated polyester, and a dried collagen (porcine) vitrigel membrane	Enhanced re-epithelialization; reduced inflammation;	[109]
Rats (Wistar)	Full-thickness skin wounds	Bovine collagen type I	Improved wound healing efficacy; enhanced fibroblast proliferation; enhanced collagen deposition and re-epithelialization; enhanced angiogenesis	[110]
Rats (Wistar)	Full-thickness skin wounds	Collagen and elastin sponge derived from salmon	Improved wound healing efficacy; enhanced fibroblast proliferation; enhanced collagen deposition and re-epithelialization; enhanced angiogenesis	[111]
Rat (Sprague-Dawley)	Full-thickness skin wounds	Chitosan membrane containing collagen I nanospheres	Improved wound healing efficacy; enhanced re-epithelialization;	[112]
Rabbits	Skin wounds (burns or ulcers)	Agar/type I collagen composite membrane	Improved wound healing efficacy	[113]

EC—ethyl cellulose; PCL—polycaprolactone; PLA—poly (lactic acid).

**Table 5 ijms-26-10636-t005:** Clinical studies on tropocollagen/collagen in scar healing.

Study	Scar Type	Intervention	Outcome	Reference
Clinical Studies				
Venous leg ulcers treated with fish tropocollagen gel in a 12-week randomized controlled trial	Venous leg ulcers	Topical fish-derived tropocollagen/collagen gel	Improved wound healing efficacy; reduced local inflammation	[115]
Venous leg ulcers with fish tropocollagen gel around the wound in a 12 week 12-week randomized controlled trial	Venous leg ulcers	Topical fish-derived tropocollagen/collagen gel	Improved wound healing efficacy; reduced pain and discomfort;	[116]
The clinical efficacy of collagen dressing on chronic wounds: A meta-analysis of 11 randomized controlled trials	Chronic wounds (e.g., diabetic foot ulcers, venous leg ulcers)	Collagen dressing vs. standard of care	Improved wound healing efficacy	[120]
The effect of collagen on wound healing in patients with burn: A randomized double-blind pilot clinical trial	Burns (20–30% total body surface area)	hydrolyzed collagen-based supplement	Improved wound healing efficacy; improved clinical parameters (serum pre-albumin, shorter hospitalization)	[121]
Acellular fish skin matrix on thin-skin graft donor sites	Patients with split-thickness skin graft donor sites (radial forearm free flap reconstructions)	The fish-skin matrix is an acellular ECM device containing fish skin proteins, lipids; collagen used as a wound covering/scaffold	Improved wound healing efficacy; reduced pain and infection rates	[122]
Treatment of diabetic foot wounds with acellular fish skin graft rich in omega-3	Patients with postsurgical diabetic foot wounds	Fish skin extracellular matrix, rich in type I collagen and omega-3 fatty acids	Improved wound healing efficacy; no adverse or immune reactions	[117]
Comparison of two ECM-based dermal substitutes: 1. Fish skin acellular dermal matrix 2. Porcine small-intestine submucosa	Full-thickness wounds in patients	Scaffolds are ECM-based, primarily composed of type I collagen, plus elastin and glycosaminoglycans	Improved wound healing efficacy; no adverse or immune reactions (fish skin dermal matrix)	[118]
Clinical trial of the temporary biosynthetic dermal skin substitute based on collagen and hyaluronic acid	Split-thickness skin graft (STSG) donor sites in humans	Topical membrane collagen and hyaluronic acid-based biosynthetic dermal skin	Accelerated wound epithelialization	[123]

## 7. Clinical Applications and Bioavailability of Collagen/TC in Scar Therapy

Wound healing often results in scarring, which represents a significant aesthetic concern. Imperfect repair processes lead to permanent and sometimes disfiguring marks. The most common causes include acne, surgical procedures, burns, and injuries. Scars in visible areas, such as the face, can negatively affect patients’ self-esteem and overall well-being.

In acne vulgaris, once inflammation subsides, fibroblasts produce collagen, but this rarely restores normal dermal architecture. Instead, atrophic scars frequently develop [125]. Conversely, excessive collagen production can give rise to hypertrophic scars or keloids [124,126,127]. Effective treatment, therefore, requires dermatological and cosmetic interventions such as chemical peels, laser therapy, microneedling, and dermal fillers [128]. Combination approaches often provide better results than single-modality treatments [129].

Modern approaches increasingly combine TC with microneedling or laser therapy to enhance its dermal penetration and stimulate endogenous collagen synthesis. These combined modalities accelerate reepithelialization and improve collagen fiber organization [77]. However, the evidence remains largely based on small-scale clinical studies, and standardized protocols are lacking, making it difficult to compare results across trials.

The therapeutic efficacy of collagen/TC strongly depends on its bioavailability and formulation. Topical preparations (gels, creams, hydrogels) primarily act on superficial layers and require permeation-enhancing techniques such as microneedling or sonophoresis to reach the dermis [7,44]. Intradermal injections deliver collagen/TC directly to the extracellular matrix, increasing biological activity [130]. Yet, no consensus exists regarding the optimal route of administration or dosing frequency.

The molecular size, origin, and stability of collagen/TC also determine its bioactivity. Marine-derived formulations offer smaller peptide chains with better penetration but reduced durability, while mammalian sources provide stronger structural integrity [30]. Temperature, pH, and enzymatic degradation further affect stability; thus, formulations often include stabilizers such as glycerol and require cold storage (2–8 °C) to maintain biofunctionality [44]. These handling requirements increase cost and complexity, limiting routine clinical use.

In summary, while collagen/TC-based therapies demonstrate encouraging effects in scar remodeling and aesthetic dermatology, current evidence is fragmented. Future studies should directly compare formulations, delivery routes, and combination protocols using standardized endpoints to define clear therapeutic recommendations.

## 8. Challenges, Standardization, and Future Perspectives of Collagen/TC in Wound and Scar Management

The implementation of collagen/TC-based therapies in wound and scar management presents a range of financial and operational challenges [131]. The production of collagen dressings involves complex manufacturing processes, including sourcing, purification, and sterilization, which contribute to higher costs compared with traditional wound care products [40]. Moreover, the requirement for refrigerated storage and specialized handling further increases expenses and may limit accessibility, particularly in resource-constrained settings [44,132]. Despite these costs, studies have highlighted the potential cost-effectiveness of collagen-containing dressings. For example, Guest et al. (2018) [133] estimated that incorporating collagen dressings into treatment protocols for non-healing venous leg ulcers could reduce management costs by 40% over six months while improving healing outcomes and quality of life. However, these economic benefits are highly context-dependent and may not translate directly to scar management, where the therapeutic endpoints differ substantially. The lack of standardized assessment criteria and inconsistent definitions of “healing success” make cost-effectiveness analyses difficult to generalize across indications.

Although collagen/TC-based treatments demonstrate considerable promise, their broader clinical adoption remains limited by several challenges. A primary concern is the lack of standardized procedures, with significant variability in product composition, concentration, and source, as well as differences in administration routes, dosing, and treatment frequency. Such inconsistencies contribute to heterogeneous clinical outcomes and complicate the development of universally accepted treatment guidelines [134]. These discrepancies also affect the comparability of study results, as many investigations use collagen preparations with unclear degrees of crosslinking or variable molecular weight distributions. This methodological heterogeneity undermines reproducibility and hinders meta-analytical evaluation of treatment efficacy.

Recent advances in material engineering have expanded the potential of collagen/TC-based biomaterials. Crosslinking modifications and incorporation of bioactive compounds enhance mechanical strength and stability, while hybrid systems integrating silver or zinc oxide nanoparticles provide antimicrobial protection and promote tissue regeneration [135,136,137,138]. Although these approaches show promise in preclinical models, clinical translation remains limited, partly due to regulatory and cost-related barriers. Three-dimensional (3D) bioprinting has also emerged as a promising avenue for producing patient-specific collagen/TC scaffolds. Such constructs can be tailored to individual wound geometry and integrated with living cells or growth factors to accelerate regeneration [139,140,141]. Yet, the scalability, reproducibility, and long-term safety of 3D-printed materials require systematic evaluation before clinical adoption. To overcome current limitations, future research should prioritize the development of standardized, well-characterized collagen/TC formulations. Randomized controlled trials (RCTs) employing validated outcome measures—such as the Patient and Observer Scar Assessment Scale (POSAS) and ECM biomarker profiles (collagen I/III ratio, decorin, fibronectin)—are essential to establish reliable evidence for clinical efficacy [136,142]. In addition, comparative studies assessing cost-effectiveness and patient-reported outcomes would strengthen the translational framework for collagen/TC therapies. In conclusion, collagen/TC represents a biologically and clinically promising platform for scar and wound management. Nevertheless, addressing issues of standardization, regulatory compliance, and economic feasibility remains crucial to ensure reproducible results and facilitate its broader clinical adoption.

Overall, collagen and TC show substantial therapeutic potential in wound healing and scar management, offering both structural and bioactive benefits. However, current evidence remains fragmented due to methodological heterogeneity, variable formulations, and the absence of standardized evaluation criteria. Future efforts should focus on developing well-characterized, clinically validated collagen/TC systems supported by rigorous randomized controlled trials to establish clear therapeutic protocols and cost-effective applications.

The development of intelligent wound dressings represents a rapidly evolving frontier in regenerative medicine, with the potential to transform current approaches to wound and scar management fundamentally. The integration of bioactive TC with diagnostic and therapeutic components enables the design of multifunctional “smart dressing” systems that not only provide passive protection but also actively support the healing cascade. TC, as a natural extracellular matrix building block, facilitates fibroblast proliferation, angiogenesis, and collagen remodeling, thereby establishing a biologically favorable environment for tissue regeneration. The incorporation of real-time diagnostic features, such as pH sensors, offers the opportunity for early detection of infection and timely therapeutic intervention. Coupled with controlled, localized antibiotic release, such systems could reduce systemic drug exposure, minimize the risk of antimicrobial resistance, and improve patient safety. Moreover, these technologies may be further enhanced through integration with telemedicine platforms and mobile applications, enabling remote monitoring of wound healing dynamics, personalized treatment adjustments, and reduced hospitalization time.

Future research should focus on optimizing the biocompatibility, sensitivity, and responsiveness of these smart dressings while also addressing challenges related to large-scale manufacturing, cost-effectiveness, and regulatory approval. Robust randomized controlled trials will be required to validate their clinical efficacy and long-term safety. If these challenges are successfully addressed, intelligent TC-based dressings may emerge as a paradigm shift in wound care, offering improved functional and aesthetic outcomes while enhancing the quality of life for patients. Figure 3 illustrates a schematic of a smart wound dressing incorporating TC.

Although numerous studies have examined the effects of collagen on wound healing and tissue regeneration, this review seeks to emphasize the significance of TC as a soluble precursor of the collagen triple helix and a fundamental constituent of the extracellular matrix. Despite its pivotal structural role, the specific functions of TC remain relatively underexplored in scientific research. Nevertheless, its widespread commercial application indicates considerable potential as a valuable tool for both future experimental investigations and translational approaches.

## Figures and Tables

**Figure 1 ijms-26-10636-f001:**
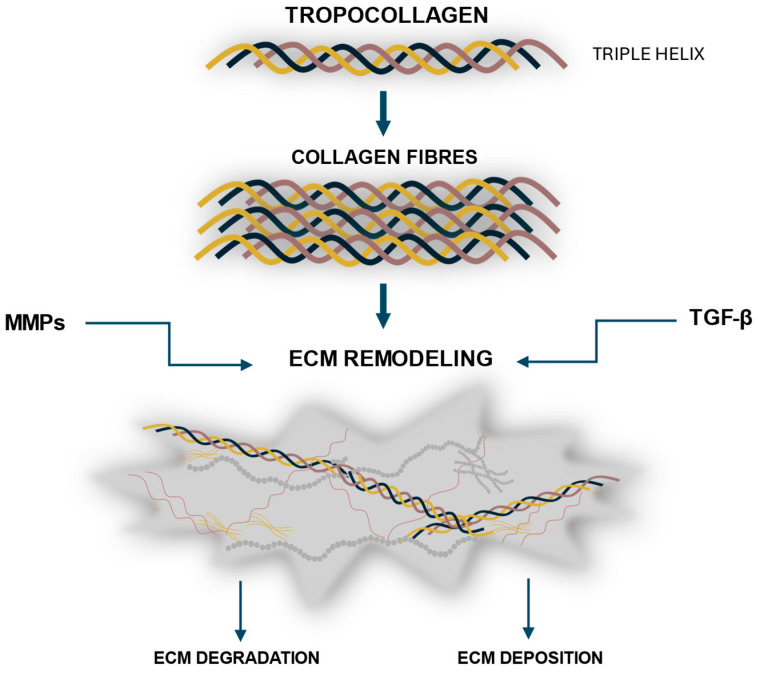
**Schematic overview of collagen formation and turnover.** Tropocollagen molecules form triple helices that self-assemble into collagen fibrils and fibers, which integrate into the extracellular matrix (ECM). The dynamic remodeling of collagen within the ECM is tightly regulated by matrix metalloproteinases (MMPs), which mediate degradation, and transforming growth factor beta (TGF-β), which promotes synthesis and organization.

**Figure 2 ijms-26-10636-f002:**
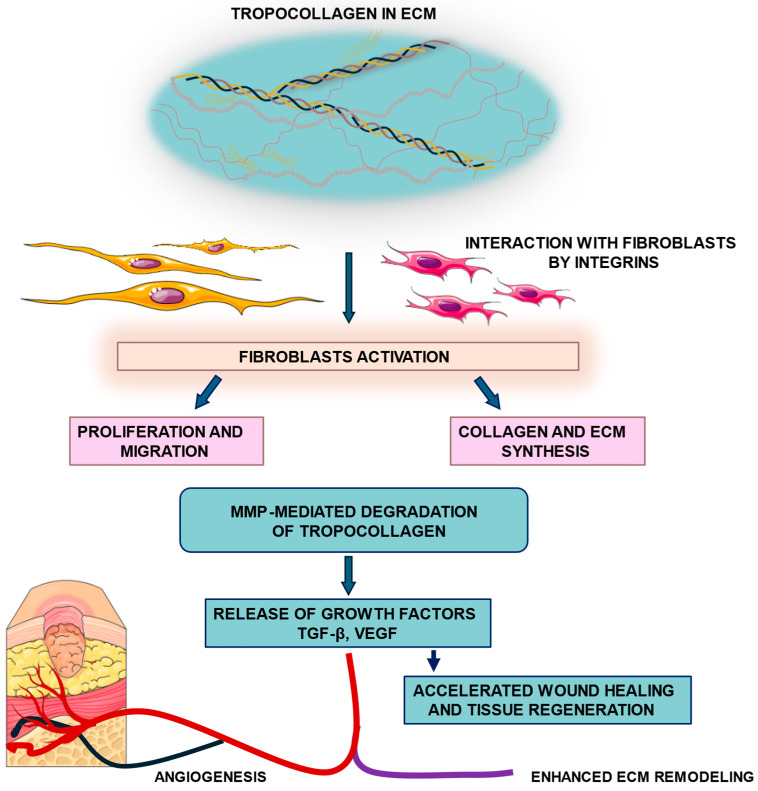
**Tropocollagen–fibroblast signaling and ECM remodeling**. Tropocollagen molecules within the extracellular matrix (ECM) interact with fibroblasts through integrin receptors, leading to fibroblast activation. Activated fibroblasts undergo proliferation and migration, and enhance collagen and ECM synthesis, contributing to matrix deposition. In parallel, matrix metalloproteinase (MMP)-mediated degradation of tropocollagen releases growth factors such as transforming growth factor beta (TGF-β) and vascular endothelial growth factor (VEGF), which enhance angiogenesis, wound healing, and tissue regeneration.

**Figure 3 ijms-26-10636-f003:**
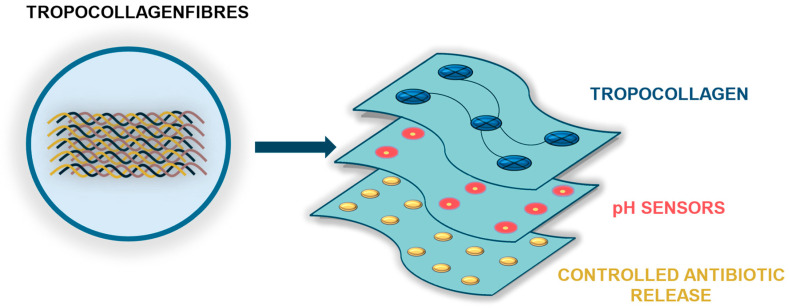
**Conceptual schematic of a smart wound dressing incorporating TC.** The dressing is composed of three functional layers: (1) a TC-based contact layer that acts as a bioactive matrix supporting fibroblast proliferation, angiogenesis, and extracellular matrix reconstruction; (2) a sensing layer for pH monitoring, enabling early infection detection and real-time wound status assessment; and (3) a therapeutic layer designed for controlled, targeted antibiotic release in response to adverse changes in the wound microenvironment.

## Data Availability

The datasets used during the current study are available from the corresponding author on reasonable request.

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
