# Peer review of "Current Insights and Future Directions in Scar Management and Skin Regeneration"

_ijms, 2025, doi:10.3390/ijms262110636_

Round 1
Reviewer 1 Report
Comments and Suggestions for Authors
The manuscript reviews recent advances in the biology and therapeutic use of tropocollagen and collagen in wound healing and scar repair. It outlines the structural features of collagen preparations, their bioactivity in extracellular matrix remodelling, and applications in regenerative dermatology. The topic is broadly relevant and references are appropriately focused on the last two years, however, some aspects of the manuscript could be refined to improve clarity, consistency, and presentation.
1- The section currently titled "Introduction" should have a more informative name.
2- Figures need full legends describing what is shown, not only titles or lists of abbreviations.
3- Section 2 should include “tropocollagen” in its title, and not only its abbreviation.
4- The title of Table 2 should specify what outcomes are being summarised (effects on what?).
5- The title of Table 3 should appear above the table, but abbreviations should be below it.
6- References in Table 3 should follow the same format as those in the rest of the manuscript.
7- Table 3 is excessively long and should be divided into three smaller tables for in vitro, in vivo, and clinical studies, in that order.
8- The writing could be made more concise and critical rather than descriptive.
9- Some sections could be merged to create a clearer distinction between biological mechanisms and therapeutic applications.
Author Response
The manuscript reviews recent advances in the biology and therapeutic use of tropocollagen and collagen in wound healing and scar repair. It outlines the structural features of collagen preparations, their bioactivity in extracellular matrix remodelling, and applications in regenerative dermatology. The topic is broadly relevant and references are appropriately focused on the last two years, however, some aspects of the manuscript could be refined to improve clarity, consistency, and presentation.
We would like to sincerely thank you for your thoughtful evaluation and constructive comments on our manuscript. We have carefully revised the text according to all your suggestions and believe that these changes have significantly improved the clarity, consistency, and overall presentation of our work. We hope that the revised version will meet your expectations and be considered positively for publication.
- The section currently titled "Introduction" should have a more informative name.
We thank the reviewer for this valuable comment, which enhances the informational clarity of the introductory section. The suggested title has been incorporated accordingly: Tropocollagen in skin regeneration and scar modulation
- Figures need full legends describing what is shown, not only titles or lists of abbreviations.
Thank you for your helpful suggestion. We have added full descriptive legends to all figures, as you recommended.
- Section 2 should include “tropocollagen” in its title, and not only its abbreviation.
We fully agree with the reviewer’s suggestion and have revised the section title to include the full term “tropocollagen.”
- The title of Table 2 should specify what outcomes are being summarised (effects on what?).
The title of Table 2 has been revised in accordance with the reviewer’s suggestion.
- The title of Table 3 should appear above the table, but abbreviations should be below it
We fully agree with the reviewer’s comment and thank them for the suggestion. The table titles and abbreviations have been corrected accordingly in all tables throughout the manuscript.
- References in Table 3 should follow the same format as those in the rest of the manuscript.
We apologize for the oversight and thank the reviewer for pointing it out. The references in Table 3 have been carefully revised to ensure consistency with the formatting used throughout the manuscript.
- Table 3 is excessively long and should be divided into three smaller tables for in vitro, in vivo, and clinical studies, in that order.
We thank the reviewer for this valuable observation. In response, we have divided the original table into three separate tables following the suggested sequence, and the corresponding descriptions in the text have been adjusted to match the new table order.
- The writing could be made more concise and critical rather than descriptive. Some sections could be merged to create a clearer distinction between biological mechanisms and therapeutic applications.
We sincerely thank the reviewer for their insightful and constructive comments, which were highly beneficial in improving the final version of our manuscript. We have carefully addressed all recommendations and revised the text accordingly, making it more concise, critical, and better structured in line with the reviewer’s suggestions.

Reviewer 2 Report
Comments and Suggestions for Authors
The authors provided a comprehensive review on the importance of collagen in tissue remodeling and a summary of current applications of collagen-related products. The authors discussed the use of tropocollagen, gelatin, collagen peptides, and native collagen fibrils in various applications such as cell culture, tissue engineering, wound closure, etc. Overall, a lot of topics were covered in this review, but it might be beneficial if more in-depth discussion and analysis could be given.
Specific comments:
- In section 4 and figures 1 & 2, the discussion on ECM remodeling and the complex interactions between fibroblasts, immune cells, MMPs, cytokines, and other growth factors are a bit too vague.
- In section 6, the authors provided a summary of various clinical and in vitro studies, but critical analysis is not presented. Moreover, the purpose of these studies and their experimental groups are so different (and some were done in conjunction with other synthetic polymers) that it was hard for any lateral comparisons to be made and therefore difficult for readers to extrapolate useful information. The outcomes of each study was only presented in a qualitative fashion instead of quantitative.
- The authors could consider offering a clearer comparison of the advantages and disadvantages of different forms of collagen, e.g. collagen peptides vs native fibrils, for various applications.
- The citation style was not consistent, some citations were done with [x] and some with (xxx et al.).
Author Response
The authors provided a comprehensive review on the importance of collagen in tissue remodeling and a summary of current applications of collagen-related products. The authors discussed the use of tropocollagen, gelatin, collagen peptides, and native collagen fibrils in various applications such as cell culture, tissue engineering, wound closure, etc. Overall, a lot of topics were covered in this review, but it might be beneficial if more in-depth discussion and analysis could be given.
We sincerely thank the Reviewer for the thorough evaluation and valuable comments. We have carefully revised the manuscript to incorporate all suggested changes and improvements. We hope that the updated version now meets the Reviewer’s expectations and provides a more in-depth and comprehensive discussion of the topic.
Specific comments:
- In section 4 and figures 1 & 2, the discussion on ECM remodeling and the complex interactions between fibroblasts, immune cells, MMPs, cytokines, and other growth factors are a bit too vague.
Thank you for this insightful comment. We have expanded the discussion in Section 4 and revised Figures 1 and 2 to provide a more detailed explanation of ECM remodeling and the complex interactions between fibroblasts, immune cells, MMPs, cytokines, and growth factors, as suggested.
- In section 6, the authors provided a summary of various clinical and in vitro studies, but a critical analysis is not presented. Moreover, the purpose of these studies and their experimental groups are so different (and some were done in conjunction with other synthetic polymers) that it was hard for any lateral comparisons to be made, and therefore difficult for readers to extrapolate useful information. The outcomes of each study were only presented in a qualitative fashion instead of a quantitative.
Thank you for this valuable comment. We fully agree with the reviewer’s assessment that the presented studies are highly diverse in terms of objectives, experimental models, and materials used, which indeed makes direct comparisons difficult. Unfortunately, we cannot address this limitation further, as the currently available data are inherently heterogeneous and fragmented. As discussed in the later sections of the manuscript, there is a clear need to redefine the research approach to collagen and tropocollagen studies to ensure more consistent and comparable outcomes in the future.
It is also challenging to perform a meaningful quantitative analysis when the tested preparations were not well characterized, the applied doses varied or were not specified, and combined experimental methods were often used. Nevertheless, we fully acknowledge the reviewer’s point and have incorporated additional critical remarks regarding the studies presented in the tables to better highlight these limitations.
We also believe it is valuable to present all the accumulated results, even if heterogeneous, as this allows us to conclude the urgent need for standardization of studies. Such standardized research could provide better insights and improved perspectives for wound and scar treatment.
- The authors could consider offering a clearer comparison of the advantages and disadvantages of different forms of collagen, e.g. collagen peptides vs native fibrils, for various applications.
The citation style was not consistent, some citations were done with [x] and some with (xxx et al.).
Thank you for these valuable suggestions. We have included a clearer comparison of the advantages and disadvantages of different forms of collagen, such as collagen peptides and native fibrils, for various applications. Additionally, we have standardized the citation style throughout the manuscript according to the journal’s requirements and your recommendation.

Round 2
Reviewer 1 Report
Comments and Suggestions for Authors
The authors have successfully addressed all my concerns.
Reviewer 2 Report
Comments and Suggestions for Authors
The authors have restructured the tables to provide more clarity. No further comments from me.